# Meta-Analysis of Mechano-Sensitive Ion Channels in Human Hearts: Chamber- and Disease-Preferential mRNA Expression

**DOI:** 10.3390/ijms241310961

**Published:** 2023-06-30

**Authors:** Elisa Darkow, Dilmurat Yusuf, Sridharan Rajamani, Rolf Backofen, Peter Kohl, Ursula Ravens, Rémi Peyronnet

**Affiliations:** 1Institute for Experimental Cardiovascular Medicine, University Heart Center Freiburg∙Bad Krozingen, 79110 Freiburg im Breisgau, Germany; 2Medical Center and Faculty of Medicine, University of Freiburg, 79110 Freiburg im Breisgau, Germany; 3Spemann Graduate School of Biology and Medicine (SGBM), University of Freiburg, 79104 Freiburg im Breisgau, Germany; 4Faculty of Biology, University of Freiburg, 79104 Freiburg im Breisgau, Germany; 5Bioinformatics Group, Department of Computer Science, University of Freiburg, 79110 Freiburg im Breisgau, Germany; 6Translational Safety and Bioanalytical Sciences, Amgen Research, Amgen Inc., South San Francisco, CA 91320, USA; 7Centre for Integrative Biological Signalling Studies (CIBSS), University of Freiburg, 79104 Freiburg im Breisgau, Germany

**Keywords:** stretch-activated ion channels, ischemic cardiomyopathy, dilated cardiomyopathy

## Abstract

The cardiac cell mechanical environment changes on a beat-by-beat basis as well as in the course of various cardiac diseases. Cells sense and respond to mechanical cues via specialized mechano-sensors initiating adaptive signaling cascades. With the aim of revealing new candidates underlying mechano-transduction relevant to cardiac diseases, we investigated mechano-sensitive ion channels (MSC) in human hearts for their chamber- and disease-preferential mRNA expression. Based on a meta-analysis of RNA sequencing studies, we compared the mRNA expression levels of MSC in human atrial and ventricular tissue samples from transplant donor hearts (no cardiac disease), and from patients in sinus rhythm (underlying diseases: heart failure, coronary artery disease, heart valve disease) or with atrial fibrillation. Our results suggest that a number of MSC genes are expressed chamber preferentially, e.g., *CHRNE* in the atria (compared to the ventricles), *TRPV4* in the right atrium (compared to the left atrium), *CACNA1B* and *KCNMB1* in the left atrium (compared to the right atrium), as well as *KCNK2* and *KCNJ2* in ventricles (compared to the atria). Furthermore, 15 MSC genes are differentially expressed in cardiac disease, out of which *SCN9A* (lower expressed in heart failure compared to donor tissue) and *KCNQ5* (lower expressed in atrial fibrillation compared to sinus rhythm) show a more than twofold difference, indicative of possible functional relevance. Thus, we provide an overview of cardiac MSC mRNA expression in the four cardiac chambers from patients with different cardiac diseases. We suggest that the observed differences in MSC mRNA expression may identify candidates involved in altered mechano-transduction in the respective diseases.

## 1. Introduction

Cardiac cells are exposed to continually changing mechanical forces leading to stretching, compression, torsion, bending, shearing or a combination of these. Cardiac cells can sense and transduce mechanical cues and modify signaling pathways accordingly. Molecules involved in the process of mechano-sensation and -transduction are integrins, various transmembrane receptors such as G-protein coupled receptors and receptor tyrosine kinases and mechano-sensitive ion channels (MSC) [1]. 

The term MSC is here considered in its broadest sense, i.e., it includes mechanically modulated channels (channels responding indirectly to mechanical stimulation or requiring co-activation) and mechanically gated ion channels (channels responding directly to mechanical stimulation). The latter comprise volume-activated ion channels and stretch-activated ion channels that activate in response to changes in membrane in-plane tension or curvature [2]. Based on their ion selectivity, stretch-activated ion channels can be subdivided into two main families: cation non-selective and potassium-selective. Additionally, there are MSC candidates, referring to ion-conducting pathways that have not yet been shown to be mechano-sensitive, but which—based on phylogenetic analysis [3] and sequence similarity [4], e.g., the number of transmembrane domains of the corresponding proteins and/or links with mechano-transduction pathways—are predicted to be mechano-sensitive. 

While mechano-transduction is important for physiological adaptive responses, excessive and/or prolonged mechanical perturbation may lead to maladaptive tissue remodeling [5]. In the heart, various transient receptor potential ion channels (TRP) that mediate stretch-induced calcium influxes have been associated with induction of fibrosis and arrhythmias [6,7,8,9], cardiac hypertrophy [10] and heart failure (HF) [11]. However, it remains unclear whether MSC remodeling occurs as a cause or a consequence of cardiac disease.

With this study, we aim to obtain a better understanding of the mRNA expression of MSC in cardiac pathophysiology and to unravel possible candidates for chamber-selective drug therapy to avoid ventricular side effects. To address this, we used previously published RNA sequencing datasets [12,13,14,15,16] including non-diseased tissue samples and high patient numbers. Our results demonstrate that human cardiac MSC are differentially expressed (mRNA) between heart chambers and in health and disease.

## 2. Results

With the aim of identifying novel MSC genes underlying pathological conditions, we conducted a meta-analysis of five previously published bulk-tissue RNA sequencing datasets. Thus, we included data from non-diseased (no structural heart disease) cardiac tissue samples and a large number of patients with dilated cardiomyopathy (DCM), ischemic cardiomyopathy (ICM), coronary artery disease (CAD), heart valve disease (HVD) or atrial fibrillation (AF). Samples originated from the left ventricle (LV), right ventricle (RV), left atrium (LA) or right atrium (RA). LV samples with DCM and ICM constitute the HF group and samples from RA with CAD and HVD without AF compose the SR group.

### 2.1. Principal Component Analysis 

When we conducted pairwise comparisons, whole genome differential gene expression between atria and ventricles explained 53% of the variance between individual non-diseased samples, whereas sex-associated differences in gene expression explained 10% of the variance (Figure 1a). In diseased LV samples, principal component 1 roughly separated samples originating from HF patients from those originating from donor hearts (explaining 22% variance; Figure 1b). In RA samples, samples originating from AF patients were interlaced with those originating from SR (Figure 1c).

### 2.2. Mechano-Sensitive Ion Channels and Their Cardiac mRNA Expression 

The mRNA expression of non-diseased and diseased human tissue from ventricles and atria was analyzed with respect to the MSC and MSC candidate genes listed in Table 1. Cardiac MSC were mainly selected based on [17,18]. Additional candidate MSC were extracted from the gene ontology resource [19,20] (GO:0008381; organism: Homo sapiens) where they have been assigned ‘mechano-sensitive’ based on phylogenetic analysis [3] and sequence similarity [4]. In total, considering MSC and candidate MSC, 73 genes were considered in this study.

We characterized cardiac mRNA expression of MSC in non-diseased samples of all heart chambers (Figure 2). The heatmap represents mRNA expression in normalized counts for each tissue sample and for each MSC gene. The dendrograms on both sides of the heatmap represent the similarity between the levels of mRNA expression. We observed a clustering of the samples by tissue provenance (i.e., depending on chamber): in no case were ventricular and atrial samples clustered together. All MSC genes are expressed in the heart, except for *ASIC5* and *KCNK4*, which clustered together because both are not systematically expressed. This cluster showed highest similarity to *ASIC2*, *GRIN1*, *TRPC5* and *KCNA1*, which are among the least-expressed MSC genes in the human heart. Furthermore, we observed two clusters of MSC genes with atria-preferential mRNA expression—on the one hand, *KCNA5* and *CHRNE* and, on the other hand, *KCNMB2*, *KCNMB1*, *KCNQ3*, *KCNMA1* and *SCN9A*. In contrast, *GJA3*, *TMC6* and *KCNJ2* may cluster for their ventricle-preferential mRNA expression. Finally, *CLCN3*, *CACNA1C* and *SCN5A* clustered together for their strong mRNA expression throughout heart chambers. Taken together, the heatmap illustrates that MSC have a wide spectrum of mRNA expression in the heart.

### 2.3. Chamber-Preferential MSC mRNA Expression 

Chamber-preferential MSC mRNA expression was analyzed using non-diseased human cardiac tissue samples. Depending on grouping of samples (i.e., atria vs. ventricles, only LA vs. LV or only RA vs. LA), we found that a total of 21 MSC displayed chamber-preferential gene expression. Of these 21, 17 were higher expressed in the atria than in the ventricles (atria-preferential). 4 of these 21 were higher expressed in the ventricles than in the atria (ventricle-preferential). An overview for all comparisons is provided in Table A1. MSC genes are considered differentially expressed when the adjusted *p*-value is lower than 0.05 (Benjamini–Hochberg procedure). Figure 3 and Figure 4 show differentially expressed genes having a |log_2_(fold difference)| ≥ 1, to increase chances to select differential expressions that are most likely to be biologically relevant.

Comparing both-sides atria to both-sides ventricles (atria vs. ventricles; Figure 3a), we found chamber-preferential gene expression for 20 MSC. A total of 16 of these 20 were atria-preferential: *CHRNE*, *KCNA5*, *ASIC4*, *KCNQ5*, *PIEZO2*, *TRPM3*, *KCNQ3*, *SCNN1A*, *KCNMB2*, *KCNA1*, *CACNA1B*, *ASIC1*, *TMC5*, *KCNMB1*, *FAM155A* and *TRPV4*. 4r of these 20 were ventricle-preferential: *KCNK2*, *KCNJ2*, *KCNJ8* and *TMC6*. At the whole genome level, *KCNA5* ranged among the top 10 differentially expressed genes, *CHRNE* among the top 20 and *KCNJ2* among the top 110 (Figure 4a). 

Comparing LA to LV (Figure 3b), we found chamber-preferential gene expression for 18 MSC; 14 of these 18 were atria-preferential. Except for *KCNA1*, *FAM155A* and *TRPV4*, all of the atria-preferential MSC genes observed in Figure 3a showed LA-preferential mRNA expression. Therefore, atria-preferential mRNA expression of *KCNA1*, *FAM155A* and *TRPV4* could be due, at least in part, to high expression in the RA. One gene, *KCNMA1*, was expressed higher in LA compared to LV, but comparing both-sides atria to both-sides ventricles, *KCNMA1* mRNA expression was not significantly different. A total of 4 of these 18 chamber-preferential MSC genes observed in Figure 3b were ventricle-preferential. All of the ventricle-preferential MSC genes observed in Figure 3a showed LV-preferential mRNA expression. At the whole genome level, *KCNA5* was the top differentially expressed gene, *KCNJ2* ranged among the top 60 differentially expressed genes and *CHRNE* among the top 80 (Figure 4b).

Comparing both atrial sides (RA vs. LA; Figure 3c), we found chamber-preferential gene expression for three MSC. *CACNA1B* and *KCNMB1* were expressed LA preferentially. *TRPV4* was expressed RA preferentially. At the whole genome level, *KCNMB1* ranged among the top 140 differentially expressed genes (Figure 4c). 

*CHRNE* atria/LA-preferential expression was the most pronounced with two orders of magnitude difference, followed by *KCNA5*, *ASIC4*, *KCNQ5* and *TRPM3* atria/LA-preferential expression with one order of magnitude difference. For the other genes with atria-preferential expression, the order of magnitude difference was < 1 for at least one comparison. None of the MSC genes with ventricle-preferential mRNA expression reached two orders of magnitude difference. *KCNJ2* ventricular/LV-preferential expression was the most pronounced with one order of magnitude difference for both comparisons. For the other genes with ventricular-preferential expression, the order of magnitude difference was <1 for at least one comparison (Table 2). 

### 2.4. Disease-Preferential MSC mRNA Expression 

To address disease-preferential MSC expression, we compared mRNA levels between diseased and control (non-diseased or SR) human cardiac tissue samples. MSC genes are considered differentially expressed when the adjusted *p*-value is lower than 0.05 (Benjamini–Hochberg procedure). In Figure 5 and Figure 6, differentially expressed genes satisfying additionally |log_2_(fold difference)| < 1 are included, and highlighted by a lighter shade of red/blue than those with a more than twofold difference in expression levels. We found that 15 MSC have a differential expression depending on the disease considered. Among those, expression of *CHRNE, KCNJ4* and *TRPC6* was higher and expression of *SCN9A*, *CFTR*, *ASIC3*, *LRRC8A*, *KCNJ11* and *TMEM63A* was lower in HF samples compared to donor hearts (Figure 5a). *SCN9A* is the only MSC gene in HF with a |log_2_(fold difference)| ≥ 1. For *TRPC6, SCN9A, LRRC8A* and *ASIC3*, HF-related differential expression seemed to be driven largely by DCM (Figure 5b). In AF, expression of *PKD1, KCNJ4* and *KCNQ4* was higher and expression of *KCNQ5, TMC5* and *KCNJ5* was lower compared to samples from patients in SR (Figure 5c). *KCNQ5* is the only MSC gene in AF with a |log_2_(fold difference)| ≥ 1. Except for *KCNJ4*, this differential mRNA expression seemed to be dominated by CAD (Figure 5d). One gene, *TMEM120A*, was expressed higher in AF than in CAD, but showed no significant difference in expression when comparing AF to SR (Figure 5c,d). 

At the whole genome level, no MSC ranged among the top 10 differentially expressed genes. *SCN9A* was within the top 200 differentially expressed genes when comparing HF vs. donor (Figure 6a). When comparing DCM vs. donor, none of the MSC genes reached a differential expression with a |log_2_(fold difference)| ≥ 1 (Figure 6b). *KCNQ5* ranged among the top 140 and the top 170 when comparing AF vs. SR and AF vs. CAD, respectively (Figure 6c,d). None of these MSC genes with disease-preferential expression exceeded one order of magnitude difference (Table 3). 

To conclude, we found that MSC have a wide range of expression in the heart, that sex was correlated with gene expression, and 21 MSC were expressed chamber-preferentially (among them, 11 MSC displayed ≥ one order of magnitude difference). From diseased tissues, 15 MSC were differentially expressed (among them only *SCN9A* [lower in HF vs. donor] and *KCNQ5* [lower in AF vs. SR or CAD] showed a |log_2_(fold difference)| ≥ 1, and none displayed ≥ 1 order of magnitude difference). 

## 3. Discussion

In this project we aimed to describe MSC mRNA expression in cardiac physiology and pathophysiology. We sought to characterise MSC mRNA expression in human cardiac health and disease with the aim of obtaining new insights on the involvement of MSC in cardiac pathophysiology. Our main results are (i) expression of acetylcholine receptor nicotinic subunit ε (*CHRNE*) shows strong preferential expression in the atria (compared to ventricles), (ii) TRPV4 (*TRPV4*) is expressed RA preferentially (compared to LA), (iii) Ca_v_2.2 (*CACNA1B*) and BK_Ca_ β-1 subunits (*KCNMB1*) are expressed LA preferentially (compared to RA), (iv) TREK-1 (*KCNK2)* and K_ir_2.1 (*KCNJ2)* are higher expressed in ventricles compared to atria, (v) Na_v_1.7 (*SCN9A*) is lower expressed in HF (compared to donor) and (vi) K_v_7.5 (*KCNQ5*) is lower expressed in AF (compared to SR or CAD).

Principal component analysis showed that differential gene expression correlates with heart chamber, sex and health status (Figure 1). Our observation that samples originating from AF did not differ clearly from those originating from SR may be explained by the fact that all patients with AF also had CAD or HVD as an indication for cardiac surgery (just as SR patients), but tissue samples were only grouped by atrial rhythm. In addition, all diseased RA samples were obtained from male patients. Thus, no sex-associated differences could be explored. We did not account for age and sex in the various sample groups because we performed a retrospective study. That means that we could not match sample sizes to the different parameters (but prioritized a higher sample size). For the same reason, we could not conduct all theoretically possible pairwise comparisons. For example, for the non-diseased samples we could not compare RA vs. RV and LV vs. RV because these samples groups were not balanced within the previously published reference datasets that we had access to.

The heatmap visualizes the normalized counts per MSC gene across all samples (Figure 2). On the one hand, it allows for an assessment of the amount of mRNA expression across heart chambers. We found that a large number of MSC are expressed in the heart, and most of them are widely present throughout the four chambers. Only *ASIC5* and *KCNK4* are not systematically expressed in the human heart (they are predominantly expressed in the intestinal tract and in the brain, respectively). On the other hand, the heat map enabled a first evaluation of chamber-preferential mRNA expression as a complementary approach to the analysis presented in Figure 3. Consequently, for *KCNA5*, *CHRNE*, *KCNMB2* and *KCNQ3*, atria-preferential and for *TMC6* and *KCNJ2* ventricle-preferential mRNA expression was identified by both approaches. With this, we confirm the well-known atria-preferential expression of *KCNA5* [70,71]). That said, we should not forget that minimally expressed MSC may still play a major role in cardiac physiology [72]. They simply may be expressed in a small subpopulation of cells, e.g., nerve cells, a subgroup of immune cells. In addition, a few channels with a large conductance, e.g., BK, may have major roles although lowly expressed.

Differentially expressed genes in non-diseased human cardiac tissue derived from different heart chambers are summarized in Table 2 (*cf.* Figure 3). Overall, 21 MSC genes of the 73 considered were differentially expressed in cardiac chambers. Comparing LA to LV, we confirmed higher *KCNA5* expression and lower *KCNJ2* expression but we could not confirm higher *SCN5A*, *SCN9A* and *KCNJ5* expression in LA when compared to LV [73,74]. Gaborit et al. also included the comparison RA vs. RV into their analysis, which showed that *KCNA5*, *KCNJ4* and *KCNJ5* are predominantly expressed in RA, and *KCNJ2* and *KCNJ8* are predominantly expressed in RV [74]. When comparing RA to LV by microarray analysis, Barth et al. found *CHRNE*, *KCNA5*, *PKD2* and *TRPC1* to be higher expressed in RA, and *KCNJ2*, *KCNJ4*, *KCNJ8*, *KCNQ4*, *SCN5A* and *TRPM3* to be predominantly expressed in LV [75].

Differentially expressed genes in cardiac tissue derived from patients with different health conditions are summarized in Table 3 (*cf.* Figure 5). Overall, 15 MSC genes of the 73 considered were dysregulated in the context of the cardiac diseases included in this study. We confirmed lower *CFTR* [76,77] and *KCNJ11* [78] expression in HF as well as higher *TRPC6* [11] expression in DCM. While we only found lower RA mRNA expression of *KCNJ5* in AF compared to SR and CAD, an earlier study showed increased *KCNJ5* expression in AF compared to controls [79]. Furthermore, we could not confirm lower ventricular expression of *KCNJ8* in HF compared to non-diseased samples [80]. When comparing ICM and DCM, Liu et al. showed that *KCNJ5* and *KCNMB1* are higher expressed in ICM [12].

Interestingly, the largest differences in MSC mRNA expression were observed when comparing chambers (donor hearts), not when considering the health status of the tissue. Indeed, only *SCN9A* and *KCNQ5* showed a |log_2_(fold difference)| ≥ 1 in disease, compared to 11 genes that showed one-to-two orders of magnitude difference when comparing chambers. Mechanical activity and constraints are very different and probably most conserved in donor hearts between atria and ventricles. Therefore, it is not surprising to have several MSC genes with a marked chamber-preferential mRNA expression. When comparing chambers, *CHRNE* ranges among the top 20 most differentially expressed genes at the whole genome level. However, very little is known about its role in the heart, making it a very exciting target to investigate further.

When considering MSC mRNA expression in the context of diseases, among the more highly expressed genes this meta-analysis identifies *PKD1.* Although it is differentially expressed in the context of AF (*p*-value < 0.05), the biological relevance of this difference |log2(fold difference)| < 1 remains to be considered. This channel is ubiquitous in mammalian cells, involved in various mechano-transduction pathways and is known to contribute to regulate Piezo1 channels [48,81], a channel also ubiquitously expressed and related to AF [82]. Although not much is known about the role of TRPP1 in the heart, it was recently reported to assemble with K_v_ channels to change cardiomyocytes repolarization and contractility [83]. Taken together, this observation suggests that TRPP1 might be an interesting mechano-sensor to investigate further in the context of the heart and particularly for AF.

The top 10 differentially expressed genes associated with a heart chamber or cardiac diseases confirm earlier findings which help to validate our study (Figure 4 and Figure 6). Although mRNA expression of the matrix metallopeptidase 3, MMP3, was strongly lower expressed in AF when comparing to SR, its low average expression (19.3 counts) suggests that it may not be biologically relevant. Of note, in the context of AF, an upregulation of *MMP3* was reported previously [84,85,86] which is contradictory to our result.

Especially for ion channels, including MSC, it may not be a change in expression but a change in activity that contributes to cardiac pathophysiology. In addition, it is difficult to correlate gene expression with the number of functional proteins in the plasma membrane. For example, in atrial myocytes, K_v_1.5 has been shown to respond to shear stress by increased insertion into the plasma membrane, i.e., recruitment from intracellular compartments [87]. Consequently, the validation of our findings at the protein level will be the subject of future research. Having said that, the necessity of “genomic approaches to identify and investigate genes associated with AF and HF susceptibility” was recently demonstrated by Patel et al. [88].

Whole tissue RNA sequencing does not allow for interpretation about the cell type responsible for differential gene expression and for quantification of heterogeneous responses in individual cell populations. This implies that large differences in expression may exist within one cell type but may not be detected in this study because the whole tissue is considered. This could be particularly significant if the gene of interest is expressed in multiple cell types but differed in only one. Most functional data were acquired in a specific cell type (often cardiomyocytes). The obtained results will be advanced by single-cell or single-nucleus RNA sequencing. Moreover, quantitative polymerase chain reaction and functional experiments like patch-clamp can be used to confirm expression and characterise candidate MSC genes. Furthermore, the genetic background was not consistent within the dataset but could not be accounted for in the analysis, in part, because the information could not be retrieved from all original studies.

This meta-analysis gives an overview of MSC mRNA expression in cardiac health and disease, and highlights MSC chamber- and disease-preferential mRNA expression, suggesting novel potential molecular targets involved in cardiac mechano-transduction.

## 4. Materials and Methods

### 4.1. Studies Considered for the RNA Sequencing Meta-Analysis

We accessed deposited transcriptomic datasets via the National Center for Biotechnology Information sequence read archive database [89]. Studies had to fulfill the following criteria for inclusion into this meta-analysis: organism: homo sapiens; tissue type: cardiac; source: RNA; strategy: RNA sequencing; platform: Illumina; library layout: paired end; file type: FASTQ. The sequencing files of human cardiac tissue samples were then grouped, based on the patients’ health status, into non-diseased (no structural heart disease), DCM, ICM, CAD, HVD or AF. Non-diseased atrial and ventricular myocardial samples originated from donor hearts not suitable for transplantation due to size mismatch or logistic reasons. Diseased myocardial samples were obtained from patients undergoing open heart surgery. All patients had given informed consent within the respective studies. Only tissue from patients with sustained (non-paroxysmal) AF was included in the AF group. Of note, all patients with AF also had CAD or HVD as the reason for cardiac surgery but tissue samples were only grouped into AF. Furthermore, the samples had to have clearly identified origins (tissue provenance) from the LV, RV, LA or RA. This meta-analysis included paired (different tissue provenances from the same individual) and non-paired tissue samples. In order to relate our results to published work, especially results of functional experiments, LV samples with DCM and ICM constitute the HF group and samples from RA with CAD and HVD without AF compose the SR group. Considering these inclusion criteria, the analysis of RNA sequencing studies with a total of 108 samples from 62 individuals is summarized in Table 4. Details on sample acquisition, including ethical approvals, and technical procedure can be found in the corresponding original publications. Patient characteristics are summarized in Table 5.

### 4.2. RNA Sequencing Data Analysis

RNA sequencing data analysis was carried out using the Galaxy platform [90], following guidelines from the tutorial “Reference-based RNA-Seq data analysis” [91,92]. In short, data were downloaded and extracted in FASTQ format from the National Center for Biotechnology Information sequence read archive [89]. Thomas et al. [15] deposited 20 sequencing runs/sample to achieve higher coverage. All runs for one sample, i.e., FASTQ files separated into forward and reverse datasets, were merged using Concatenate datasets tail-to-head [93]. Quality control checks on raw sequence data were performed using FastQC [94] and MultiQC [95]. Then, Cutadapt [96] was used to remove adapter sequences. The splice-aware aligner STAR [97] was used to map the RNA sequencing reads onto the human reference genome (hg19). Mapping results were visualized by the integrative genome viewer, IGV [98]. Thereafter, the strandness of the RNA sequencing data (reads mapping to the forward or reverse DNA strand) was estimated using Infer Experiment from RSeQC [99]. Gene expression was measured by featureCounts [100].

From the published reference datasets, we selected and combined the samples into three different groups of pairwise comparisons where the condition samples were balanced in each respective group: AF vs. SR (from RA tissue samples), HF vs. non-diseased (from LV tissue samples) and atria vs. ventricles (from non-diseased tissue samples). For the comparison of AF with SR, we selected 25 samples from the reference datasets of Thomas et al. [15] and Darkow et al. [16]. To compare HF with non-diseased, we selected 28 samples from the reference datasets of Darkow et al. [16], Schiano et al. [13] and Liu et al. [12]. To compare atria with ventricles, we selected 53 samples from the reference datasets of Johnson et al. [14] and Darkow et al. [16]. 

For each pairwise comparison, we used the DESeq2 package [101] in R [102] to test for differential expression of genes, and employed the gglot2 package [103] to visualize the overall effect of experimental covariates with the plots of principal component analysis. To perform the differential gene expression analysis in DESeq2, we utilized the formula “design~batch + condition” to model the gene expression data according to the experimental design. This formula represents the relationship between the gene expression data and the experimental conditions, while also accounting for potential batch effects. The formula incorporates “design” which specifies how samples are categorized. The “~” symbol signifies the relationship between variables in the R formula syntax. Including “batch” corrects technical variations derived from the different references, while “condition” represents the health status or the tissue provenance. By integrating both “batch” and “condition,” the formula “design ~ batch + condition” allows for accurate identification of differentially expressed genes, mitigating technical artifacts. The resulting “normalized” counts were transformed by the function vst() of DESeq2 and further underwent batch removal with the function removeBatchEffect() of the limma package [104], which served as the input to generate the principal component analysis plots. For checking differential expression, we first ran the function DESeq(), and then applied the function lfcShrink() with the shrinkage estimator ashr [105]. The analysis parameters and the versions of the tools and packages used until this step can be found here: https://dyusuf.github.io/analysis_collection/Peyronnet-lab/Darkow2023_data_analysis.html. 

Differential gene expression levels with adjusted *p*-values < 0.05 (Benjamini–Hochberg procedure) were regarded as significantly different (statistical significance); the condition |log_2_(fold difference)| ≥ 1 was regarded as indicative functional relevance (biological significance); data representation was performed with OriginPro 2020. DESeq2 output files were re-uploaded to the Galaxy platform and used to generate a heatmap and volcano plots (Wickham, 2009) [103].

## Figures and Tables

**Figure 1 ijms-24-10961-f001:**
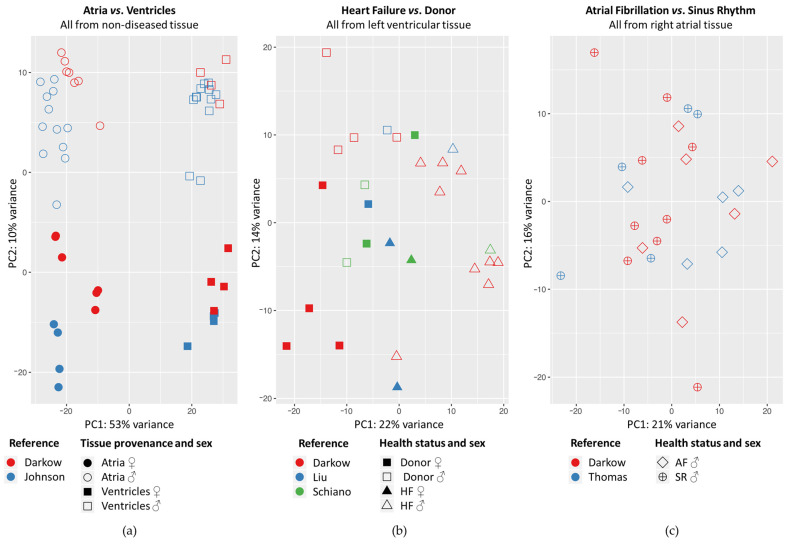
Principal component (PC) analysis illustrating inter- and intra-group variance of samples per comparison: (**a**) atria (including left and right atrium) vs. ventricles (including left and right ventricle) in non-diseased (donor) tissue; (**b**) heart failure (HF; including dilated and ischemic cardiomyopathy) vs. donor in left ventricular tissue; (**c**) atrial fibrillation (AF) vs. sinus rhythm (SR; including coronary artery and heart valve disease) in right atrial tissue. ♂: male patient/donor; ♀: female patient/donor; each individual point represents data from one tissue sample; the whole genome was considered.

**Figure 2 ijms-24-10961-f002:**
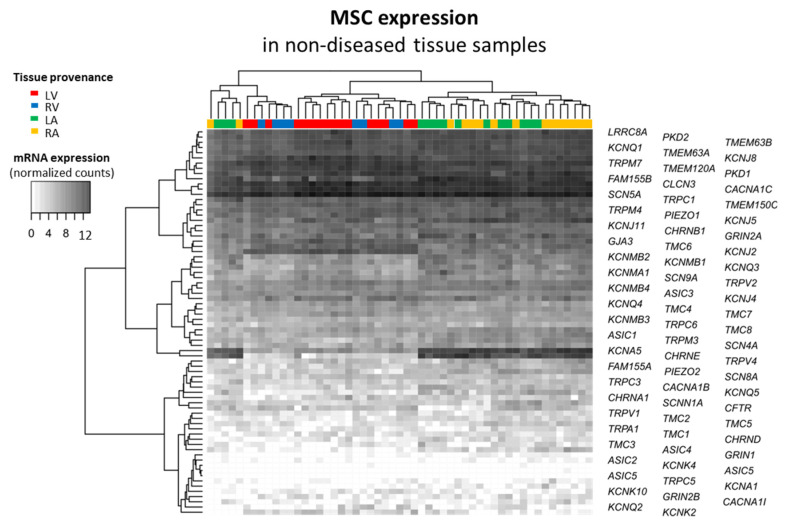
MSC mRNA expression in non-diseased human cardiac tissue samples. Gene expression assessed by RNA sequencing; expressed in normalized counts derived from the comparison atria vs. ventricles; tissue provenance: left ventricle (LV), right ventricle (RV), left atrium (LA) and RA; each column represents data from one tissue sample; each line represents one MSC gene. The dendrograms were obtained with the Euclidean distance method and the complete hierarchical clustering method.

**Figure 3 ijms-24-10961-f003:**
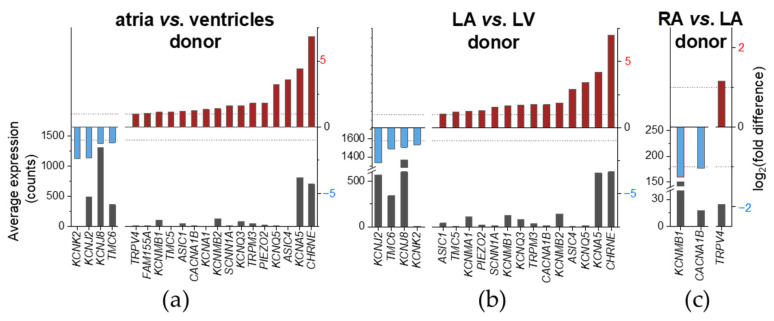
Chamber-preferential MSC mRNA expression in human cardiac tissue samples. Left ordinate indicates the average mRNA expression over all samples from both conditions expressed in normalized counts; right ordinate indicates the log_2_ (fold difference) of the comparison. MSC genes significantly (adjusted *p*-value < 0.05; Benjamini–Hochberg procedure) higher (left on the abscissa, red) or lower (right on the abscissa, blue) expressed are shown; unit example: log_2_(fold difference) = 2 🡢 fold difference = 4. Dashed lines indicate |log_2_(fold difference)| = 1. Atrial (pooled from right [RA] and left [LA] atrium) compared to ventricular (pooled from right [RV] and left [LV] ventricle; (**a**)), LA compared to LV (**b**), and RA compared to LA (**c**) non-diseased tissue samples.

**Figure 4 ijms-24-10961-f004:**
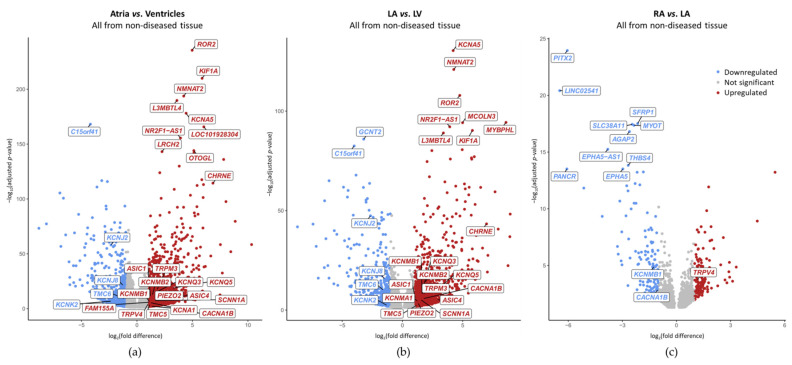
Chamber-preferential mRNA expression in non-diseased human cardiac tissue samples. Each dot represents one gene; among the genes with adjusted *p*-value < 0.05, MSC and top 10 differentially expressed genes are labelled (none of the top 10 are MSC). Genes with |log_2_(fold difference)| ≥ 1 are color-coded: higher (red) or lower (blue) expression (**a**) in the atria than in the ventricles; (**b**) in the left atrium (LA) than in the left ventricle (LV); (**c**) in the right atrium (RA) than in LA.

**Figure 5 ijms-24-10961-f005:**
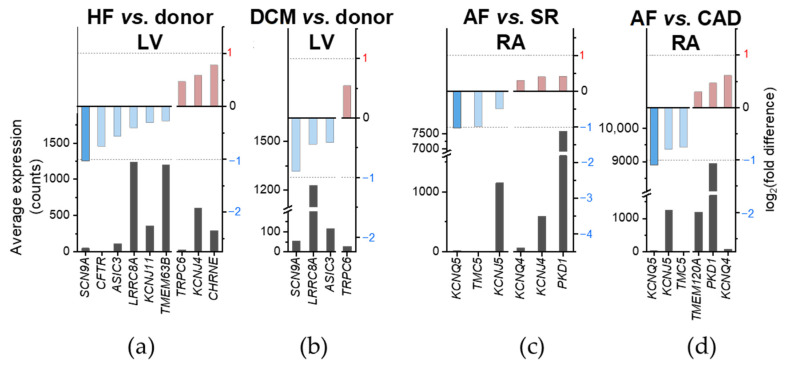
Disease-preferential MSC mRNA expression in human cardiac tissue samples. Left ordinate indicates the average mRNA expression over all samples from both conditions expressed in normalized counts; right ordinate indicates the log_2_(fold difference) of the comparison. MSC genes significantly (adjusted *p*-value < 0.05; Benjamini–Hochberg procedure) higher (left on the abscissa, red) or lower (right on the abscissa, blue) expressed are shown. Lighter shade of red/blue indicate |log_2_(fold difference)| < 1. (**a**,**b**) Heart failure (HF; pooled from dilated cardiomyopathy [DCM] and ischemic cardiomyopathy; (**a**)) and DCM (**b**) compared to non-diseased LV samples. (**c**,**d**) Atrial fibrillation (AF) compared to sinus rhythm (SR; pooled from coronary artery disease [CAD] and heart valve disease; (**c**)) and compared to CAD (**d**) diseased RA samples.

**Figure 6 ijms-24-10961-f006:**
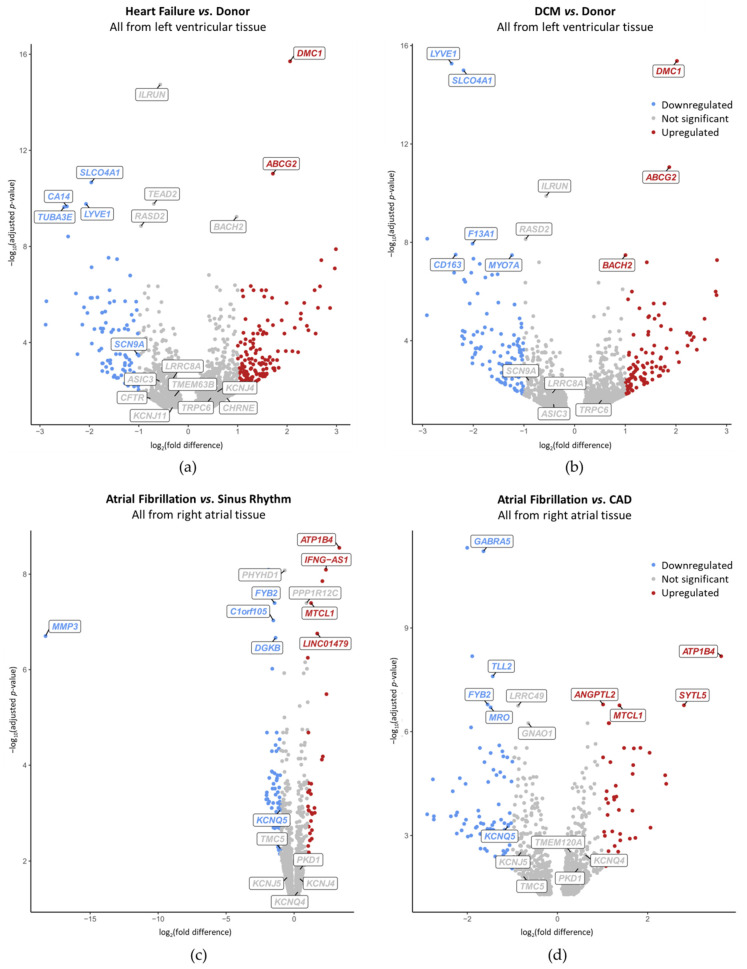
Disease-preferential mRNA expression in left ventricular (**a**,**b**) and right atrial (**c**,**d**) human cardiac tissue samples. Each dot represents one gene; among the genes with adjusted *p*-value < 0.05, MSC and top 10 differentially expressed genes are labelled. Genes higher expressed than |log_2_(fold difference)| ≥ 1 (red) or lower (blue) are colored. (**a**,**b**) Heart failure (HF; pooled from dilated cardiomyopathy [DCM] and ischemic cardiomyopathy; (**a**)) and DCM (**b**) compared to non-diseased donor hearts. (**c**,**d**) Atrial fibrillation (AF) compared to sinus rhythm (SR; pooled from coronary artery disease [CAD] and heart valve disease; (**c**)) and compared to CAD (**d**).

**Table 1 ijms-24-10961-t001:** Human mechano-sensitive ion channels (MSC) and candidates in alphabetical order. Gene and protein names are shown together with references demonstrating their direct or indirect involvement in mechano-sensing. When mechano-sensitivity has been attributed to specific subunits only these were selected—if not, all subunits were considered. MMCs, mechanically modulated ion channels; VACs, volume-activated ion channels; SACs, stretch-activated ion channels; SAC_K_, K^+^-selective SACs; SAC_NS_, cation non-selective SACs; Ca^2+^, calcium; Na^+^, sodium; K^+^, potassium; H^+^, hydrogen; Cl^−^, chloride.

Gene	Protein	Category	Reference
*ASIC1,2,3,4,5*	Acid-sensing ion channel 1,2,3,4,5	MMC (Na^+^)	[21]
*CACNA1B,C,I*	Voltage-dependent N,L,T-type Ca^2+^ channel subunit α-1B,C,I or Ca_v_2.2, 1.2, 3.3	MMC (Ca^2+^)	[22,23,24]
*CFTR*	Cystic fibrosis transmembrane conductance receptor	MMC (Cl^−^)	[25]
*CHRNA1,B1,D,E*	Acetylcholine receptor nicotinic subunit α1,β1,δ,ε	MMC	[26]
*CLCN3*	H^+^/Cl^-^ exchange transporter 3 or CLC-3	MMC (Cl^−^)	[27]
*FAM155A,B*	Transmembrane protein FAM155A,B	candidate	[3]
*GJA3*	Gap junction α-3 protein or Connexin 46	MMC	[28]
*GRIN1, 2(A,B)*	Ionotropic glutamate receptor NMDA type subunit 1, 2		[29,30]
*KCNA1,5*	K^+^ voltage-gated channel subfamily A member 1,5 or K_v_1.1,1.5	SAC_K_	[31,32]
*KCNJ2,4*	Inward rectifier K^+^ channel 2,4, or K_ir_2.1,2.3		[33,34]
*KCNJ5*	G protein-activated inward rectifier K^+^ channel 4 or K_ir_3.4 or GIRK4		[35]
*KCNJ8,11*	ATP-sensitive inward rectifier K^+^ channel 8,11 or K_ir_6.1,6.2		[36]
*KCNK2,4,10*	K^+^ channel subfamily K member 2,4,10 or TREK-1, TRAAK, TREK-2		[37,38,39]
*KCNMA1*	Ca^2+^-activated K^+^ channel subunit α-1 or K_Ca_1.1 or BK_Ca_ α	SAC_K_, MMC (K^+^)	[40]
*KCNMB1,2,3,4*	Ca^2+^-activated K^+^ channel subunit β-1,2,3,4 or BKβ1,2,3,4		
*KCNQ1,2,3,4,5*	K^+^ voltage-gated channel subfamily KQT member 1,2,3,4,5 or K_v_7.1,7.2,7.3,7.4,7.5	VAC	[41,42,43,44]
*LRRC8A*	Leucine rich repeat containing 8 VRAC subunit A or SWELL1		[45,46]
*PIEZO1,2*	Piezo-type mechano-sensitive ion channel component 1,2	SAC_NS_	[47]
*PKD1,2*	Polycystin-1,2 or Transient receptor potential cation channel subfamily P member 1,2 or TRPP1,2	MMC (Ca^2+^)	[48,49]
*SCN(4,5,8,9)A*	Na^+^ channel protein type 4,5,8,9 subunit α or Na_v_1.4,1.5,1.6,1.7	SAC_NS_, MMC (Na^+^)	[50,51,52,53]
*SCNN1A*	Na^+^ channel epithelial 1 subunit α or ENaCα	SAC_NS_	[54]
*TMC1,2,3,4,5,6,7,8*	Transmembrane channel like protein 1,2,3,4,5,6,7,8	MMC (Cl^−^), candidate	[3,55]
*TMEM120A*	TACAN	SAC_NS_	[56]
*TMEM150C*	Transmembrane protein 150C or Tentonin 3	MMC	[57]
*TMEM63A,B*	CSC-1 like protein 1,2	SAC_NS_, candidate	[4,58]
*TRPA1*	Transient receptor potential cation channel subfamily A member 1	SAC_NS_	[59]
*TRPC1,3,5,6*	Short transient receptor potential channel 1,3,5,6		[60,61,62,63]
*TRPM3,4,7*	Transient receptor potential cation channel subfamily M member 3,4,7		[64,65,66]
*TRPV1,2,4*	Transient receptor potential cation channel subfamily V member 1,2,4		[67,68,69]

**Table 2 ijms-24-10961-t002:** Summary of chamber-preferential MSC expression in non-diseased human cardiac tissue samples. Arrows indicate the general direction and the orders of magnitude of the difference in expression. Arrows indicate higher or lower expression, no significant difference is indicated by (−): significant differences < 1 order of magnitude (↗ or ↘), significant differences of 1 order of magnitude or more (↑ or ↓); alphabetical order. LA, left atrium; LV, left ventricle; RA, right atrium; LA, left atrium.

Gene	Atria vs. Ventricles	LA vs. LV	RA vs. LA
*ASIC1*	↗	↗	−
*ASIC4*	↑	↑	−
*CACNA1B*	↗	↑	↘
*CHRNE*	↑	↑	−
*FAM155A*	↗	−	−
*KCNA1*	↗	−	−
*KCNA5*	↑	↑	−
*KCNJ2*	↓	↓	−
*KCNJ8*	↘	↘	−
*KCNK2*	↓	↘	−
*KCNMA1*	−	↗	−
*KCNMB1*	↗	↗	↘
*KCNMB2*	↗	↑	−
*KCNQ3*	↗	↑	−
*KCNQ5*	↑	↑	−
*PIEZO2*	↑	↗	−
*SCNN1A*	↗	↗	−
*TMC5*	↗	↗	−
*TMC6*	↘	↘	−
*TRPM3*	↑	↑	−
*TRPV4*	↗	−	↗

**Table 3 ijms-24-10961-t003:** Summary of disease-preferential MSC expression. Arrows indicate the general direction and the orders of magnitude of the difference in expression. Arrows indicate higher or lower expression, no significant difference is indicated by (−): significant differences < 1 order of magnitude (↗ or ↘), significant differences of 1 order of magnitude; alphabetical order. HF, heart failure; DCM, dilative cardiomyopathy; AF, (sustained) atrial fibrillation; SR, sinus rhythm; CAD, coronary artery disease.

Gene	HF vs. Donor	DCM vs. Donor	AF vs. SR	AF vs. CAD
*ASIC3*	↘	↘	−	−
*CFTR*	↘	−	−	−
*CHRNE*	↗	−	−	−
*KCNJ4*	↗	−	↗	−
*KCNJ5*	−	−	↘	↘
*KCNJ11*	↘	−	−	−
*KCNQ4*	−	−	↗	↗
*KCNQ5*	−	−	↘	↘
*LRRC8A*	↘	↘	−	−
*PKD1*	−	−	↗	↗
*SCN9A*	↘	↘	−	−
*TMC5*	−	−	↘	↘
*TMEM120A*	−	−	−	↗
*TMEM63B*	↘	−	−	−
*TRPC6*	↗	↗	−	−

**Table 4 ijms-24-10961-t004:** Studies considered for the meta-analysis. SRA, sequence read archive; HF, heart failure; DCM, dilated cardiomyopathy; ICM, ischemic cardiomyopathy; SR, sinus rhythm; CAD, coronary artery disease; HVD, heart valve disease; AF, sustained atrial fibrillation; LV, left ventricle; RV, right ventricle; LA, left atrium; RA, right atrium; N, number of individuals (donors or patients); n, number of tissue samples.

Reference	SRA Accession Number	Platform	Health Status		Tissue Provenance	Structural Heart Disease	Sample Size (n/N)
[12]	PRJNA246308	Illumina HiSeq 2000	Non-diseased		LV	No	2/2
			HF	DCM ICM	LV	Yes	2/2
							1/1
[13]	PRJNA291619	Illumina HiSeq 2000	Non-diseased		LV	No	4/4
			HF	DCM	LV	Yes	2/2
[14]	PRJNA445706	Illumina HiSeq 2500	Non-diseased		LV RV LA RA	No	32/8
[15]	PRJNA526687	Illumina NextSeq 500	SR	CAD	LA, RA	Yes	8/4
				HVD	RA		1/1
			AF		LA RA	Yes	10/5
[16]	PRJEB42485	Illumina HiSeq 4000	Non-diseased		LV	No	21/8
					LA		
					RA		
			HF	DCM	LV	Yes	5/5
				ICM			4/4
			SR	CAD	RA	Yes	6/6
				HVD			4/4
			AF		RA	Yes	6/6

**Table 5 ijms-24-10961-t005:** Clinical data of patients included in the RNA sequencing meta-analysis. DCM, dilated cardiomyopathy; ICM, ischemic cardiomyopathy; CAD, coronary artery disease; HVD, heart valve disease; AF, atrial fibrillation; LV, left ventricle; RV, right ventricle; LA, left atrium; RA, right atrium; SEM, standard error to the (arithmetic) mean.

Tissue Provenance	LV			RV	LA			RA			
Health Status	Non- Diseased	DCM	ICM	Non- Diseased	Non- Diseased	CAD	AF	Non- Diseased	CAD	HVD	AF
Sample size	22	9	5	8	14	4	5	15	10	5	11
Sex (♂/♀)	13/9	7/2	4/1	6/2	9/5	4/0	5/0	10/5	10/0	5/0	11/0
Age (mean ± SEM)	51.2 ± 2.6	52.6 ± 3.8	54.2 ± 4.1	47.4 ± 4.4	53.2 ± 3.2	62.5 ± 4.0	73.4 ± 2.4	54.3 ± 3.0	64.2 ± 3.1	53.4 ± 7.2	70.2 ± 1.6

## Data Availability

The datasets presented in this study can be found in online repositories. The names of the repositories and accession numbers can be found below: https://www.ncbi.nlm.nih.gov/geo/query/acc.cgi?acc=GSE57345; https://www.ncbi.nlm.nih.gov/geo/query/acc.cgi?acc=GSE71613; https://www.ncbi.nlm.nih.gov/geo/query/acc.cgi?acc=GSE112339; https://www.ncbi.nlm.nih.gov/geo/query/acc.cgi?acc=GSE128188; https://www.ebi.ac.uk/ena/browser/view/PRJEB42485 (all accessed on 7 September 2021).

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
