# Peer review of "Meta-Analysis of Mechano-Sensitive Ion Channels in Human Hearts: Chamber- and Disease-Preferential mRNA Expression"

_ijms, 2023, doi:10.3390/ijms241310961_

Round 1

Reviewer 1 Report

In the manuscript titled “Meta-Analysis of Mechano-Sensitive Ion Channels in Human Hearts: Chamber- and Disease-Preferential mRNA Expression”, Elisa Darkow and team sought to analyze mechano-sensitive ion channels (MSCs) in human hearts to explore chamber and disease-specific mRNA expression using RNA-seq data gathered from public repositories. The group found there to be differences based on chambers, and certain genes are differentially expressed in various cardiac diseases. The manuscript is well-written and highlights some interesting findings, with a concise abstract that gives a nice overview. I also found the tables and figures to be well-organized and helpful. However, I have a few concerns that I note below.

Major concerns:

1.      I appreciate that you included batch and condition as covariates in your models to account for the different references, however there continues to be prominent separations in the PC1 vs PC2 plots; have you tried other functions such as combat?

2.      Did you adjust for age and sex?

3.      I am confused about why you specifically chose 21 of the 73 that were ultimately noted to be chamber-preferential. Is this what you found when doing pairwise comparisons for all of the chambers? Also, I only noted exploration of atria vs ventricles, left vs right atrium, and left atrium vs left ventricle. What about right atrium vs right ventricle and left vs right ventricle?

4.      Please include and expand on the limitations of the study more clearly.

5.      My biggest concern is the lack of validation in your findings—I am not sure if you plan to measure expression in in vivo (i.e., mouse) or in vitro models, but how do you plan to address this?

6.      There is a complete absence of female samples among coronary artery disease, atrial fibrillation and heart valve disease, and only 3 of 14 cardiomyopathy samples are females. Have you done any sex-stratified analyses among the non-diseased samples? Have you done sensitivity analyses excluding the females from the cardiomyopathy samples to explore for any changes to your findings?

Minor concerns:

1.      Not a concern, just commenting that I appreciate the information provided in Tables 1, 2 and especially 4.

2.      Table 3 needs more organization—perhaps dividing into cation-nonselective and potassium-selective, and channel families and isoforms.

3.      Please expand on the Discussion section.

Author Response

Please accept our apologies for sending the wrong report, Please find below replies to your comments:

Major concerns:

  1. I appreciate that you included batch and condition as covariates in your models to account for the different references, however there continues to be prominent separations in the PC1 vs PC2 plots; have you tried other functions such as combat?

We thank you for highlighting this point. When performing differential expression analysis, it is important to distinguish between modelling a batch effect and directly modifying data to counteract batch effects. Programs like ComBat aim to remove batch effects by modifying the data, so that statistical tests can be conducted without factoring in batch. In contrast, including 'batch' as a covariate in a design formula models the effect size of the batch, without altering the data. This modelled effect size is subsequently considered in the statistical inferences, which is approach we would favour. 

To enable descriptive analyses through PCA plots, we utilized limma::removeBatchEffect on counts after variance stabilizing transformations. This approach followed the Deseq2 guidelines, effectively removing batch effects. The separation between Johnson and Darkow among female atrial sample data in Figure 1a along PC2 accounts for 10% of the variance dominated by sex. However, no reference-driven separations were observed along PC2 in the remaining samples. The same can be said regarding PC1 in all samples, which explains 53% of the variance driven by tissue provenance. This shows that 'batch' is not a predominant factor responsible for the separation in PCA. Moreover, in this type of analysis, it is often observed that a small cluster of samples of the same batch shows up due to biological effects. This observation is evident in Figure 4 of the ComBat publication: https://academic.oup.com/nargab/article/2/3/lqaa078/5909519.

  1. Did you adjust for age and sex?

This comment relates closely to comment 6 and we will address both here. We did not account for sex or age while conducting differential expression analysis, for the reasons outlined below.

Performing a descriptive analysis with PCA is recommended before determining the necessity of adjustment. This enables assessing the effect of sex or age on contrasting biological conditions. 

Regarding the effect of sex on atria vs. ventricles, Figure 1a exhibits a sex-related effect along PC2. However, it is only present among samples of either atria or ventricles, but not between (i.e. atria vs. ventricles), hence it would appear to not have an impact on the contrast between the two. Instead of performing sex-stratified analyses, as suggested in comment 6, we computed the differential expression between atria and ventricles while accounting for sex by using the design formula of ‘batch + sex + condition’. We also measured the correlation between the differential expression of MSC via adjusting and non-adjusting for sex (‘batch + condition’). The adjusted version shows no significant difference compared to the non-adjusted one, as shown in Suppl. Figure 1 where the correlation coefficient is 1.

Please see Suppl. Figure 1 (attached)

In Figure 1b, comparing heart failure to donors, the sex-related effect is not immediately clear. However, as you note in Comment 6, there is an imbalance in sample size between conditions, highlighting a common issue in meta-analyses. We conducted a comparison between unadjusted and sex-adjusted results, and Suppl. Figure 2 shows that the correlation coefficient between the two is 0.98. Consequently, the adjustment would appear to have a negligible effect on outcome.

Please see Suppl. Figure 2 (attached)

Age is not a categorical variable, which is why meaningful age groups were established to be used for modelling. However, upon examining the descriptive analyses, we could not discern the age group effect across atria vs. ventricles (Suppl. Figure 3), heart Failure vs. donor (Suppl. Figure 4) or atrial Fibrillation vs. sinus rhythm (Suppl. Figure 5). Besides these findings, people in the same age group are likely to have differing patho/physiological characteristics. As a result, we do not have any evidence to suggest that adjusting for age groups would enhance our analysis of differential expression in these datasets.

Please see Suppl. Figure 3, 4 and 5 attached.

  1. I am confused about why you specifically chose 21 of the 73 that were ultimately noted to be chamber-preferential. Is this what you found when doing pairwise comparisons for all of the chambers? Also, I only noted exploration of atria vs ventricles, left vs right atrium, and left atrium vs left ventricle. What about right atrium vs right ventricle and left vs right ventricle?

Here, we use “chamber-preferential mRNA expression” for genes that display differential expression in either atria or ventricles. We found 21 MSC genes that are either higher or lower expressed in the atria (right or left or both) or the ventricles (right or left or both). They correspond to the genes displayed in the new figure 3, panel a – c. 20 of them appear in panel a (atria vs. ventricles; broadest classification). Panel b (left atrium vs. left ventricle) and c (right atrium vs. left atrium) show the same genes, except for KCNMA1 which is the 21st gene that we identified as being expressed chamber preferentially. In the latter panels we were able to assign chamber-preferential expression to specific heart chambers, although not exhaustively. The reason is that we could not conduct pairwise comparisons for right atrium vs. right ventricle and left vs. right ventricle, because among our selected references there were not two that included samples with the same tissue provenance and the same health condition. We added this to the limitations (L258).

  1. Please include and expand on the limitations of the study more clearly.

We expanded the limitations on the aspect of different age and sex distribution within the different samples groups; we mentioned that our pairwise comparisons are non-exhaustive, and that our findings need to be validated at the protein level (L258 and 349).

  1. My biggest concern is the lack of validation in your findings—I am not sure if you plan to measure expression in in vivo (i.e., mouse) or in vitro models, but how do you plan to address this?

We agree that validation of our findings at the protein level is the next logical step. However, the scope of this study was to give an overview of MSC mRNA expression in the heart, to guide conceptual development and further experimental research. Indeed, we found interesting candidates which we intend to explore (pending funding). We added this to the discussion (L349).

  1. There is a complete absence of female samples among coronary artery disease, atrial fibrillation and heart valve disease, and only 3 of 14 cardiomyopathy samples are females. Have you done any sex-stratified analyses among the non-diseased samples? Have you done sensitivity analyses excluding the females from the cardiomyopathy samples to explore for any changes to your findings?

Please see our response under comment 2, above.

Minor concerns:

  1. Not a concern, just commenting that I appreciate the information provided in Tables 1, 2 and especially 4.

Thank you very much.

  1. Table 3 needs more organization—perhaps dividing into cation-nonselective and potassium-selective, and channel families and isoforms.

We appreciate your suggestion. However, we suggest to not change the table. The reasons are: i) we do not refer to the classes of MSC elsewhere in the manuscript, ii) if we go by classes, readers would need to know the class of MSC if they wanted to look up a specific MSC. We suspect some readers won’t know all classes of MSC and they might face difficulties to find their MSC of interest. iii) Some classes are not very well defined or quite wide (e.g. cation nonselective stretch-activated channels includes many channels)..

  1. Please expand on the Discussion section.

We extended the discussion on the heatmap (starting L258) and we expanded the study limitations section (L258 and 349). We also extended the comparison of our results with other previously published data.

Reviewer 2 Report

I would like to thank the authors for performing the valuable research and writing it up, and the editor for giving me the opportunity to review the resulting paper.

The topic of the paper is mechano-sensitive ion channels (MSC) which function as mechano-sensors in heart cells. The authors study mRNA expression levels of MSC in tissue samples from different parts of hearts that were either healthy or had different diseases. They observe differences both between parts of the heart and between diseases, and speculate that these differences are important in the disease mechanisms. RNA-sequencing results for 108 samples from 62 individuals obtained in several earlier studies are used in the meta-analysis. 73 MSC and candidate MSC genes are considered. For subsets of samples, PCA is performed on the gene expression data. For example, for non-diseased tissue, the first component explains 53 % of the variance and beautifully separates the samples into atria and ventricles, while the second component explains 10 % of variance and splits the samples by gender. Similar comparisons are made for other selections of tissues, although the first two components never explain as much variance or separate the samples so nicely. Comparisons in expression of particular genes (not only MSC) between subsets of samples are made as well, identifying many chamber- and disease-preferential MSC genes. The main identified examples of preferential gene expression, the biological relevance of the studied genes, and the limitations of the available data are then discussed. It is also noted that whole tissue RNA-sequencing does not allow the identification of the responsible cell type or may miss important differences in expression that are limited to a certain cell type and could be identified through single-cell RNA-sequencing.

The overlap of my expertise with the contents of the article is very small and does not allow me to point out any major issues. The only remarks that I'd share and might be reasonable are:
- I find the lines 133–136 quite unclear. It is challenging to understand what the "design ~ batch + condition" model is, and the phrase "DESeq2 function" probably refers to a function from DESeq2 package? An editing mishap might have happened as well, introducing a paragraph break within a sentence.
- Is explanation for the dendrograms in Figure 1 provided? I couldn't find it.
- Do you use the 73 genes from Figure 1 in performing PCA or all the genes from Suppl. Figure 1 and 2 (by the way, how many are there and what are they)?
- The Instructions for Authors on https://www.mdpi.com/journal/ijms/instructions#suppmaterials state: "For work where novel computer code was developed, authors should release the code either by depositing in a recognized, public repository such as GitHub or uploading as supplementary information to the publication. The name, version, corporation and location information for all software used should be clearly indicated. Please include all the parameters used to run software/programs analyses." I'm not sure if you have automated the analysis to the point that anyone could run it if you shared your code. However, if someone attempted to replicate your study, a more detailed description of your procedure would be really helpful to them. Please list the versions (and other metadata) of the software mentioned in section 2.2. Consider sharing your R scripts and configuration files with the parameters, and if some steps of the analysis were performed manually, describe them as precisely as possible. (For example, when you removed the adapter sequences using Cutadapt, did you have to specify any parameters in addition to providing the results of the previous processing step? Is there a parameter file, can you provide your command-line call, or have you done it interactively, documenting your inputs? In the preceding step of quality control, did FastQC and MultiQC require parameter files, command-line parameters, and/or interactive input? Did you have to specify some quality threshold, and what did you do with the data not meeting the threshold? Similarly for featureCounts: how did you configure it? Et cetera.)

Author Response

We thank you for your evaluation of our work. Please find below our answers to your comments:

- I find the lines 133–136 quite unclear. It is challenging to understand what the "design ~ batch + condition" model is, and the phrase "DESeq2 function" probably refers to a function from DESeq2 package? An editing mishap might have happened as well, introducing a paragraph break within a sentence.

We thank you for highlighting this point. We corrected the line break and we have added a new description to the “RNA-sequencing data analysis” section for clarity (line (L) 431).

- Is expl:anation for the dendrograms in Figure 1 provided? I couldn't find it.

This was an oversight by us. We added to the figure legend how the dendrograms were obtained and we extended the figure’s description in the text as well (L127).

- Do you use the 73 genes from Figure 1 in performing PCA or all the genes from Suppl. Figure 1 and 2 (by the way, how many are there and what are they)?

The principal component analysis was based on the whole-genome. We have now clarified this and included this information in the description above the figure (L88) and also in the figure legend. Furthermore, we changed the sequence of figures, so that the principal component analysis would come before the heatmap.

The volcano plots are also based on the whole genome. In the legend of both plots, we now write: “…among the genes with adjusted p-value < 0.05, MSC and the top 10 differentially expressed genes are labelled. Genes significantly (log2(fold difference) > |1|) higher (red) or lower (blue) expressed…” Meaning that every dot represents a gene with an adjusted p-value < 0.05. Among those, only dots with a log2(fold difference) > |1| are coloured. All dots representing a MSC are labelled no matter whether the dot is coloured or not, i.e. no matter whether the genes is significantly higher or lower expressed. Finally, the top 10 differentially expressed dots/genes (of the whole genome) are also labelled.

- The Instructions for Authors on https://www.mdpi.com/journal/ijms/instructions#suppmaterials state: "For work where novel computer code was developed, authors should release the code either by depositing in a recognized, public repository such as GitHub or uploading as supplementary information to the publication. The name, version, corporation and location information for all software used should be clearly indicated. Please include all the parameters used to run software/programs analyses." I'm not sure if you have automated the analysis to the point that anyone could run it if you shared your code. However, if someone attempted to replicate your study, a more detailed description of your procedure would be really helpful to them. Please list the versions (and other metadata) of the software mentioned in section 2.2. Consider sharing your R scripts and configuration files with the parameters, and if some steps of the analysis were performed manually, describe them as precisely as possible. (For example, when you removed the adapter sequences using Cutadapt, did you have to specify any parameters in addition to providing the results of the previous processing step? Is there a parameter file, can you provide your command-line call, or have you done it interactively, documenting your inputs? In the preceding step of quality control, did FastQC and MultiQC require parameter files, command-line parameters, and/or interactive input? Did you have to specify some quality threshold, and what did you do with the data not meeting the threshold? Similarly for featureCounts: how did you configure it? Et cetera.)

We added the URL of workflows (Galaxy and R script) to the method section (L446). This contains all necessary detail, including analysis versions and analysis parameters.

Reviewer 3 Report

The manuscript by Darkow and collaborators presents a thorough meta-analysis of the transcriptomic data available from their lab and other publicly available on NCBI repository in order to find specific biomarkers of each heart chamber in health and diseases.
The analysis protocol is very well described and the data are overall well presented. However, I would rather suggest that the authors present all their data in the main manuscript. In fact the 2 supplementary figures S1 and S2 are important to be shown in the main text in order for a reader to better understand the discussion.
On the other hand, Figure 1 is barely discussed and in the legend there is an inversion between the rows and the columns labels. What are the "philogenetic" trees representing on both sides? On which parameter are they based? similarity?
It is remarkable that the data collected are biased towards the male samples and it is highly appreciable that the authors have taken the time to explain this bias.
The conclusions also include the limitation of the study, which is very honest and informative.
I wold rather suggest some minor modifications, mostly regarding the formatting.
In all the tables, please add the abbreviations after the title, not as footnote. It will improve the readability.
In Table 2, in the last row it would be better to write the mean +/- SED on one line instead of two
Table 3, it would be more informative to classify the genes by classes and then use the alphabetical order within each class of mechanotransducers. This will also help in follwing the discussion and the graphs in figure 3, for example.
In figure 3 not all the genes described in paragraph 3.3 are present (especially those between lines 238-242), at least not in all the panels which the reader is supposed to compare.
Finally, in lines 266-271 please use the gene names, at least between parenthesis, as in the rest of the manuscript, otherwise there can be some confusion.

Author Response

We thank you for your evaluation of our work. Please find below our answers to your comments:

The analysis protocol is very well described and the data are overall well presented. However, I would rather suggest that the authors present all their data in the main manuscript. In fact the 2 supplementary figures S1 and S2 are important to be shown in the main text in order for a reader to better understand the discussion.

We agree and have moved the volcano plots to the main text.

On the other hand, Figure 1 is barely discussed and in the legend there is an inversion between the rows and the columns labels. What are the "philogenetic" trees representing on both sides? On which parameter are they based? similarity?

We corrected the figure legend and added how the dendrograms were obtained. The dendrograms represent the similarity between the levels of mRNA expression ─ on the one hand for the samples and on the other hand for the MSC genes. Furthermore, we extended the discussion on the heatmap (L265).

It is remarkable that the data collected are biased towards the male samples and it is highly appreciable that the authors have taken the time to explain this bias. The conclusions also include the limitation of the study, which is very honest and informative. I wold rather suggest some minor modifications, mostly regarding the formatting. In all the tables, please add the abbreviations after the title, not as footnote. It will improve the readability.

We adjusted all tables accordingly.

In Table 2, in the last row it would be better to write the mean +/- SED on one line instead of two

We adjusted the table accordingly.

Table 3, it would be more informative to classify the genes by classes and then use the alphabetical order within each class of mechanotransducers. This will also help in follwing the discussion and the graphs in figure 3, for example.

We appreciate your suggestion. However, we suggest to not change the table. The reasons are: i) we do not refer to the classes of MSC elsewhere in the manuscript, ii) if we go by classes, readers would need to know the class of MSC if they wanted to look up a specific MSC. We suspect some readers won’t know all classes of MSC and they might face difficulties to find their MSC of interest. iii) Some classes are not very well defined or quite wide (i.e. cation nonselective stretch-activated channels includes many channels).

In figure 3 not all the genes described in paragraph 3.3 are present (especially those between lines 238-242), at least not in all the panels which the reader is supposed to compare.

We rechecked the text describing this figure. We introduced an additional line break to clarify where the text related to this figure ends, and where new content describing the volcano plot starts.

Finally, in lines 266-271 please use the gene names, at least between parenthesis, as in the rest of the manuscript, otherwise there can be some confusion.

We adjusted the indicated lines accordingly.